# A Recombinant Multivalent Vaccine (rCpa1) Induces Protection for C57BL/6 and HLA Transgenic Mice against Pulmonary Infection with Both Species of *Coccidioides*

**DOI:** 10.3390/vaccines12010067

**Published:** 2024-01-09

**Authors:** Althea Campuzano, Komali Devi Pentakota, Yu-Rou Liao, Hao Zhang, Nathan P. Wiederhold, Gary R. Ostroff, Chiung-Yu Hung

**Affiliations:** 1Department of Molecular Microbiology and Immunology, The University of Texas at San Antonio, San Antonio, TX 78249, USA; althea.campuzano@utsa.edu (A.C.); hao.zhang@utsa.edu (H.Z.); 2Department of Pathology, Graduate School of Biomedical Sciences, UT Health, San Antonio, TX 78229, USA; wiederholdn@uthscsa.edu; 3Program in Molecular Medicine, UMass Chan Medical School, Worcester, MA 01655, USA; gary.ostroff@umassmed.edu

**Keywords:** coccidioidomycosis, vaccine, cross-protection, T-cell immunity, Valley fever, HLA-DR4 mice

## Abstract

Coccidioidomycosis is caused by *Coccidioides posadasii* (*Cp*) and *Coccidioides immitis* (*Ci*), which have a 4–5% difference in their genomic sequences. There is an urgent need to develop a human vaccine against both species. A previously created recombinant antigen (rCpa1) that contains multiple peptides derived from *Cp* isolate C735 is protective against the autologous isolate. The focus of this study is to evaluate cross-protective efficacy and immune correlates by the rCpa1-based vaccine against both species of *Coccidioides*. DNA sequence analyses of the homologous genes for the rCpa1 antigen were conducted for 39 and 17 clinical isolates of *Cp* and *Ci*, respectively. Protective efficacy and vaccine-induced immunity were evaluated for both C57BL/6 and human HLA-DR4 transgenic mice against five highly virulent isolates of *Cp* and *Ci*. There are total of seven amino acid substitutions in the rCpa1 antigen between *Cp* and *Ci*. Both C57BL/6 and HLA-DR4 mice that were vaccinated with an rCpa1 vaccine had a significant reduction of fungal burden and increased numbers of IFN-γ- and IL-17-producing CD4+ T cells in the first 2 weeks post challenge. These data suggest that rCpa1 has cross-protection activity against *Cp* and *Ci* pulmonary infection through activation of early Th1 and Th17 responses.

## 1. Introduction

Coccidioidomycosis, also known as San Joaquin Valley fever, results from two phylogenetically related fungal pathogens, *Coccidioides immitis* (*C. immitis*, or *Ci*) and *C. posadasii* (*Cp*), endemic to the desert in the southwestern United States and aerial regions of Central and South America [1]. *Coccidioides* species are dimorphic fungi, growing as molds in the soil and differentiating into multinucleated spherules (~100 µm in diameter) in mammals [2]. Coccidioidomycosis typically begins as a pulmonary infection via inhalation of airborne arthrospores produced by soil-dwelling mycelia. Arthrospores then undergo isotropic growth, converting into spherules that form small compartments via septation. Each compartment contains 2–4 nuclei and subsequently develops into endospores that are then released during the endosporulation process (~300–800 endospores per spherule). Coccidioidal endospores (2–7 µm in diameter) can be extrapulmonary disseminated to the skin, bones, central nervous system, as well as other organs via the blood and lymphatic system. Therefore, chronic pulmonary and disseminated coccidioidomycosis patients may require lifelong antifungal therapies [3].

Clinical and laboratory experiments demonstrated the similarity in virulence and dimorphic lifestyle of these fungi [4]. Population genomics analysis revealed hybridization and genetic introgression between *C. posadasii* (*Cp*) and *C. immitis* (*Ci*), primarily from *Cp* to *Ci* [5]. Diverse clinical isolates exhibit hybrid genomic sequences [5,6,7]. Annotated orthologous genes shared between these two species present over 90% protein sequence identity [1,8]. However, serological tests using coccidioidal complement fixation (CF) or tube precipitin (TP) antigens cannot distinguish between *Cp* and *Ci* [4,9,10].

Knowledge gaps in coccidioidomycosis diagnosis, treatment, immunological responses, morbidity, and economic healthcare burden among increased incidence have raised great concerns [11]. Vaccination against coccidioidomycosis seems feasible, since a second infection is extraordinarily rare [12]. Our effort focuses on the development of a subunit vaccine composed of coccidioidal antigens plus an adjuvant delivery system [1,13,14]. Successful vaccine candidates should confer protection against both *Coccidioides* species. This approach requires identifying conserved antigens among clinical isolates of these two fungal species that stimulate long-lasting, antigen-specific adaptive immune responses. Several studies from our group have demonstrated that a multivalent vaccine can induce large amounts of specific T-cell clones, becoming more effective compared with an individual antigen [15,16,17]. A recombinant chimeric polypeptide antigen (rCpa1) is genetically engineered to link together the most immunogenic fragment of Ag2/Pra, as well as the full lengths of Cs-Ag and Pmp1, to five promiscuous T-cell epitopes that have high affinity to human MHC class II molecules in a single polypeptide construct, as shown in Figure 1A [18,19]. The five epitopes are derived from *C. posadasii*-specific aspartyl protease (Pep1), a-mannosidase (Amn1), and phospholipase B (Plb) antigens [15,16,18,20,21]. The multivalent rCpa1 antigen is then loaded into an adjuvant made of glucan-chitin particles to form the GCP-rCpa1 vaccine [18,19]. The vaccine has been reported to elicit a mixed CD4 T-cell mediated type I (Th1) and Th17 immunity and confers protection against an autologous isolate (C735) of *Cp* in murine models compared with a mock-immunized group (GCP-MSA) [18,19].

Our previous studies have been limited to evaluating the vaccine efficacy of GCP-rCpa1 against a single *C. posadasii* isolate. Since the two species of *Coccidioides* are highly conserved, we hypothesize that the synthetically engineered vaccine has the potential to elicit protection against coccidioidomycosis. Consequently, our current objective is to determine the potential of GCP-rCpa1 to provide protection against other clinical isolates of *C. posadasii* as well as *C. immitis* using C57BL/6 and human HLA-DR4 transgenic mouse models. These results suggest that it is feasible to develop a comprehensive *Coccidioides* vaccine, which could be significant to human health as its geographic distribution continues to expand. These studies have a potential advancement in preventing and mitigating the impact of coccidioidomycosis.

## 2. Materials and Methods

### 2.1. Fungal Culture

*Coccidioides posadasii* isolates of clinical origin (C735, Silveira, and RMSCC 3488) and *C. immitis* isolates (RS and RMSCC 2394) were all obtained from patients with confirmed coccidioidomycosis, and maintained in a BSL-3 laboratory at the University of Texas at San Antonio (UTSA) [8,22]. *Coccidioides* were grown on GYE agar plates (1% glucose, 0.5% yeast extract, and 1.5% agar) at 30 °C for 2–4 weeks to produce spores, as previously reported [23].

### 2.2. DNA Sequence Analysis

Primer 3™ software (version 4.1.0) was used to design primers for amplification of six rCpa1 antigen gene constructs (GenBank accession AVH85517.1), as described in Appendix A. Genomic DNA samples were isolated from *Coccidioides* cultures or the Fungus Testing Laboratory at UT Health San Antonio. The targeted genes were amplified using a standard PCR protocol with high-fidelity Taq polymerase (Life Technologies, Carlsbad, CA, USA; Cat# 11304011). The PCR products were sequenced and analyzed using BioEdit™ software (version 7.2). The translated aa sequences of these 6 antigens were aligned using ClustalW multiple sequence alignment software (GenomeNet, Kyoto University Bioinformatics Center, Kyoto, Japan; version 2.1).

### 2.3. Mice

All animal experiments were conducted following NIH guidelines and in compliance with the PHS Policy for Humane Care and Use of Laboratory Animals. Protocol #MU004 was approved via IACUC at UTSA. Male and female <8-week-old HLA-DR4 (DRB1A*01:01; DRB1*04:01) transgenic mice in a C57BL/6 genetic background were bred in house [24]. WT C57BL/6 mice were purchased from the NCI/CRL. One group of mice was used for intracellular cytokine staining, while another group of mice was used for fungal burden analysis. Mice were euthanized humanely by an overdose of isoflurane inhalation followed by cervical dislocation. Tissues and organs were collected post mortem.

### 2.4. Vaccination Protocol, Animal Challenge, and Evaluation of Protection

The rCpa1 construct was loaded into glucan-chitin particles (GCPs), as previously reported [18,19]. Each dose of vaccine contained 10 μg rCpa1, 200 µg yeast tRNA and 25 μg mouse serum albumin (MSA) as a trapping matrix, and 200 μg of GCPs in 200 µL PBS. Mice were subcutaneously immunized 3 times in the abdominal region at 2-week intervals followed by an intranasal challenge with 90–150 viable spores prepared from *C. posadasii* (C735, Silveira, and 3488) or *C. immitis* (RS and 2394) in 35 µL of PBS at 3 weeks after the final vaccination, as described previously [19]. GCP-MSA contained the same components of the vaccine, except rCpa1 was used as an adjuvant control. C57Bl/6 mice were sacrificed at 14 days post challenge, while the more susceptible HLA-DR4 mice were sacrificed at 9 days post challenge for determining fungal burden, as previously described [19,25].

### 2.5. Flow Cytometry Analysis

A separate group of mice was used exclusively for flow cytometry analysis, and pulmonary leukocytes were isolated, as previously reported [19,25]. Whole-lung homogenates from each vaccination condition group (unvaccinated, GCP-MSA, or GCP-rCpa1 (*n* = 5 per group)) for each timepoint (7 or 14 DPC) were homogenized into single-cell suspensions passed through a 70-μm cell strainer and spun down, and then RBCs were removed using Gibco AKC cell lysis buffer (Cat# A1049201) according to manufacturer recommendations. Pulmonary cells were filtered through a 40-μm cell strainer (Falcon), spun down, then enumerated using a hemocytometer. An aliquot of cells (5 × 10^5^) was labeled with a viability dye (BioLegend Zombie Aqua (San Diego, CA, USA); 1:1000; Cat# 423101) and fluorochrome-conjugated antibodies for enumerating activated T cells. The antibody cocktail included APC-Fire750-CD45 (clone: 30-F11, 0.25 μg), BV421-CD44 (clone: IM7, 0.25 μg), BV711-TCR-β (clone: H57-597, 0.5 μg), FITC-CD4 (clone: GK1.5, 0.25 μg), and PeCy-7-CD8b (clone: YTS156.7.7, 0.25 μg) per 10^6^ cells in 100μL FACS buffer. IFN-γ- and IL-17a-producing T cells were determined by intracellular cytokine assays, as previously described [19,23]. Cells were co-stimulated with a cocktail of anti-CD3 (BD Bioscience (Franklin Lakes, NJ, USA), Cat# BD553057) and anti-CD28 (BD Cat# BD553294) with GolgiPlug (BD Cat# BD555029) in complete RPMI for 4 h at 37 °C, 5% CO_2_. Cells were then permeabilized using Cytofix/Cytopem (BD Cat# 554722) according to manufacturer recommendations, then washed and labeled with the described cocktail (CD45, TCR-β, CD4, CD8) plus APC-IFN-γ (clone: XMG1.2, 0.9 μg), PE-IL-17a (clone: TC11-18H10.1, 0.25 μg), and BV421-IL-5 (clone: TRFK5, 0.25 μg). Cells were fixed using 2% paraformaldehyde in PBS (Alfa Aesar (Ward Hill, MA, USA), Cat# J61899). Data were acquired by a BD LSRII cytometer and analyzed using FlowJo software version 10.9.

### 2.6. Statistical Analyses

Fungal burden (CFUs) between the two groups was analyzed via the Mann–Whitney rank sum test [23,26]. Ordinary one-way ANOVA was used to analyze percent weight change, and Kruskal–Wallis analysis was used to analyze fungal burden and total cell numbers of cytokine-producing T cells in the lungs [19,23]. A *p* value of equal or less than 0.05 was considered statistically significant. Error bars represent the ±SD for percent weight change and total cell number flow cytometry analysis. Asterisks and daggers indicate statistically significant differences between GCP-rCpa1-vaccinated versus nonvaccinated mice (*) and the vaccinated versus mock groups (†). Data were analyzed using Prism 10.1.0.

## 3. Results

### 3.1. Variants of Vaccine Antigens among Clinical Isolates of Coccidioides

The multivalent rCpa1 antigen construct (GenBank: AVH85517.1; 586 aa in length) is derived from *Cp* isolate C735. We have generated a synthetic construct derived from *C. immitis* (rCpa2, GenBank: WAB54687.1; 586 aa in length). Antigens rCpa1 and rCpa2 are made up of three *Coccidioides* antigens (Ag2/Pra_27–132_, Cs-Ag_138–283,_ and Pmp1_288–454_) and five human MHC II-binding peptides derived from Pep1, Amn1, and Plb antigens [18]. Alignment sequence analysis revealed seven aa substitutions in the rCpa2 construct among the orthologs of *C. posadasii* (*Cp*) and *C. immitis* (*Ci*) (Table 1, Appendix A). Six of the seven aa substitutions are located on the whole-length antigens (Ag2/Pra, Cs-Ag, and Pmp1). The other substitution is located on the human Plb-P6 epitope. Only four to six coccidioidal orthologs of each antigen in the rCpa1 construct were available in GenBank. Therefore, we validated these differences by sequence analysis of the PCR amplicons using the gene-specific primers listed in Appendix A for an additional 39 and 17 clinical isolates that were typed to be *Cp* or *Ci*, respectively. Our results revealed that the translated aa sequences of these antigens were identical among the 39 isolates of *Cp*. Similarly, these antigens were indistinguishable among the 17 isolates of *Ci*. Overall, these data have validated that these seven aa substitutions exist between *Cp* and *Ci* (Table 1).

### 3.2. The GCP-rCpa1 Vaccine Is Protective against Multiple Clinical Isolates of Cp

Population genomic analysis of *Coccidioides* revealed extensive genome hybridization among clinical isolates of *Cp* and *Ci* [5,27]. Isolates C735 and Silveira pose typical *Cp* reference genomes, while RMSCC 3488 is a hybrid, with most of the *Cp* DNA sequence (personal communication with Dr. Bridget Baker). All three *Cp* isolates have identical protein sequences in the rCpa1 antigen [18]. First, we evaluated the protective efficacy of the GCP-rCpa1 vaccine (as illustrated in Figure 1A) against a pulmonary challenge with one of the three *Cp* isolates in a susceptible C57BL/6 murine model.

A group of mice was subcutaneously vaccinated and intranasally challenged with a lethal dose of *Coccidioides* spores, as described in Figure 1B. Unvaccinated mice (square) or mice injected with GCP adjuvant loaded with mouse serum albumin (GCP-MSA, triangle) served as negative and mock controls, respectively [18,19]. Mice were then left to rest for three weeks prior to the challenge to avoid nonspecific response by innate immunity to the vaccine. Mice vaccinated with the GCP-rCpa1 vaccine and then challenged with one of the three selected *Cp* isolates maintained their bodyweight during the first 14 days post challenge, while unvaccinated and mock (GCP-MSA) mice had significantly reduced bodyweight (Figure 1C). The decline in the weight of unvaccinated and mock mice correlated with a significant increase in fungal burden of their lungs and spleens compared with their vaccinated mouse counterparts (*, *p* < 0.05), as well as in mock-immunized compared with vaccinated mice (†, *p* < 0.05) (Figure 1D). Notably, the GCP-MSA mock mice had less reduced bodyweight, which concurred with significantly reduced fungal burden after challenge with *Cp* 3488 spores, compared with unvaccinated mice (*, *p* < 0.05). These data suggest that GCP-MSA could stimulate partially protective immunity against *Cp* 3488 isolates. However, protection via mock vaccination was not observed in the other *C. posadasii* strains tested. Therefore, the GCP-rCpa1 vaccine confers immune protection for all three tested *Cp* isolates.

### 3.3. GCP-rCpa1 Vaccine Cross-Protected C57BL/6 Mice against Ci Isolates

Amino acid substitutions may contribute to differences in antigenicity; therefore, we evaluated the protective efficacy of a GCP-rCpa1 vaccine using C57BL/6 mice that were challenged with *Ci* isolates. RS isolate has a reference genome of *Ci*, while RMSCC 2394 is a *Cp* and *Ci* hybrid with *Ci*-type protein sequences in its vaccine antigens. Mice were vaccinated and challenged with the same protocol (Figure 1B). *Cp*-C735 served as a reference isolate. GCP-MSA-immunized (triangle) and nonvaccinated mice (square) succumbed to coccidioidomycosis and significantly reduced bodyweight after a pulmonary challenge with the two tested *Ci* isolates, despite a slower rate compared with those infected with *Cp*-C735 isolate (Figure 2A). Overall, mice vaccinated with GCP-rCpa1 (circle) and separately challenged with a potentially lethal dose of spores isolated from either *Ci*-RS or *Ci*-2394 isolates had significantly reduced fungal burden in their lungs and spleen compared with their unvaccinated (*, *p* < 0.05) and mock GCP-MSA-injected counterparts (†, *p* < 0.05), like those challenged with C735 isolate (Figure 2B). Overall, our data suggest that the GCP-rCpa1 vaccine is cross-protective against both *Cp* and *Ci* species.

### 3.4. Vaccination with GCP-rCpa1 Resulted in Early Induction of a Mixed Th1 and Th17 Response but Not Th2 by Both Cp and Ci Infection

Studies have demonstrated that protective efficacy against coccidioidomycosis is associated with an early response via T helper cell expansion, particularly in Th1 and Th17 cells [18,19,23,28]. We then compared pulmonary Th-cell profiles in the lungs of C57Bl/6 mice after challenge with the autologous isolate *Cp*-C735 and other isolates of *Cp* and *Ci* at 7 and 14 DPC. Gating strategies for subsets of total CD4^+^ Th cells (Th1, Th17, and Th2) were based on the differential expression of CD45^+^, CD4^+^, CD8^+^, IFN-γ^+^, IL-17a^+^, and IL-5^+^, as shown in Figure 3A. We noted no significant expression of Th2-type cytokine between immunized and nonimmunized cohorts.

Both Th1 and Th17 cells were significantly elevated for GCP-rCpa1-vaccinated and *Cp*-C735 challenged mice compared with their unvaccinated and mock counterparts at 7 and 14 DPC (Figure 3B; *, *p* < 0.05 for vaccinated versus mock and †, *p* < 0.05 for vaccinated versus unvaccinated). Interestingly, GCP-rCpa1-vaccinated C57BL/6 mice had only significantly elevated Th17 cells at 7 and 14 DPC after challenge with *Cp-*Silveira spores (Figure 3B; *, *p* < 0.05). Furthermore, mice that were challenged with *Cp-*3488 had significantly increased amounts of Th17 cells at 14 DPC (Figure 3D).

Subsequently, we profiled the pulmonary CD4^+^ Th cells of C57Bl/6 mice that were challenged with *Ci* isolates, as shown in Figure 3A. Results revealed that numbers of both Th1 and Th17 were significantly elevated in GCP-rCpa1-vaccinated mice that were separately challenged with *Ci*-RS and *Ci*-2394 isolates at 7 DPC (Figure 4A,B). These data were compared with the cytokine profiles of mice challenged with *Cp*-C735 (*, *p* < 0.05) (Figure 3B). Although we noted an increase of Th1 cells in the GCP-MSA control mice at 14 DPC post challenge with *Ci-*RS isolate, the increase did not correlate with fungal clearance (Figure 2B). Therefore, elevated Th1 and Th17 levels in the lungs of the GCP-rCpa1-vaccinated mice during the first 14 DPC are associated with dampened fungal burden, despite the kinetics of these Th responses to a pulmonary challenge with the variable isolates not being identical.

### 3.5. GCP-rCpa1 Vaccine Conferred Protection for HLA-DR4 Transgenic Mice against Both Cp and Ci

We have reported that HLA-DR4 mice are highly susceptible to pulmonary challenge with *Cp*-C735 spores. Approximately 30% of HLA-DR4 mice can approach moribund at around 11 ± 2 days post challenge. Nevertheless, vaccination with the live attenuated (ΔT) and GCP-rCpa1 vaccines have greatly reduced fungal burden in the lungs of HLA-DR4 mice at 9–14 DPC [29]. In this study, we evaluated protective efficacy and immune responses of the GCP-rCpa1 vaccine against *Cp*-3488 and *Ci*-2394 at 9 DPC. HLA-DR4 mice challenged with *Cp*-C735 served as a reference. All three isolates (*Cp*-C735, *Cp*-3488, and *Ci*-2394) are highly virulent in the C57BL/6 model of coccidioidomycosis (Figure 1 and Figure 2). HLA-DR4 mice underwent the same vaccination regimen, as described in Figure 1B. All three groups of vaccinated mice that were separately challenged with one of the tested isolates had significantly reduced pulmonary fungal burden at 9 DPC compared with unvaccinated and mock mice (Figure 5A; *, *p* < 0.05). As observed in C57BL/6 mice, GCP-MSA adjuvant provided partial protection for HLA-DR4 mice against *Cp*-3488 isolate, as the mock mice had significantly reduced fungal burden in their lungs compared with unvaccinated mice (Figure 5A). Furthermore, both Th1 and Th17 cells were significantly elevated in the mice vaccinated against *Cp*-C735 and *Cp*-3488 compared with unvaccinated and mock counterparts (Figure 5B,C). There was a trend of increased Th17 cells in the lungs of vaccinated mice against *Ci*-2394 infection compared with those of mock mice, but not significant (Figure 5D). These data suggest that the GCP-rCpa1 vaccine confers immune protection for HLA-DR4 mice against *Cp* and *Ci* by activating a mixed Th1 and Th17 response (†, *p* < 0.05).

## 4. Discussion

Given the projected rise of coccidioidomycosis incidence and an underreporting of illness, developing a vaccine is crucial for effective infection prevention. Previous attempts for vaccine development include a formalin-killed spherule vaccine tested in 1980s clinical trials. Although FKS appeared protective in murine studies, the vaccine elicited high reactogenicity at the immunization site, and there was little difference in protection between the FKS-vaccinated group compared with placebo [30]. Two genetically engineered live-attenuated vaccine candidates (the Δ*cps1* mutant and the Δ*cts2*/Δ*cts3*/Δ*ard1* mutant, known as the ΔT mutant) have been evaluated for protective efficacy and vaccine-induced protective immunity against pulmonary coccidioidomycosis in mice [26,31]. Recently, Δ*cps1* vaccination and subsequent boosting significantly decreased fungal burden in a canine model compared with a control group [32]. The Δ*cps1* lacks the expression of a single virulence factor, and the ΔT mutant harbored deletion of two chitinase genes (*cts2* and *cts3*) and an upstream gene of *cts*3 annotated as an arabinotol-2-dehydrogenase (*ard1*). Both mutants can produce arthroconidia used as vaccines but cannot undergo endosporulation to complete a parasitic lifestyle in a chemically defined culture medium. Although these two vaccines can confer long-lasting protective immunity, concerns for genetic stability, the potential for a reversion to virulent pathogens, and safety for immunocompromised individuals have not been illustrated [33,34]. The advantage of chemically defined subunit vaccines is that they are comprised of adjuvant delivery systems and microbe-specific proteins or synthetic peptides. Our group has developed a synthetic recombinant subunit vaccine candidate to prevent coccidioidomycosis. In order to determine the safety and efficacy of the recombinant GCP-rCpa1 vaccine, we have performed extensive characterization, albeit with limitations that are addressed in this manuscript.

We first defined protective antigens in silico and in vivo, which led to synthesizing a recombinant polypeptide antigen construct, rCpa1 (NCBI Accession: KY883768) [17,18,21,35]. These antigens were selected because they are highly expressed during different stages of the parasitic cycle and elicit protective immune responses in murine models. The current rCpa1 construct derives from the *C. posadasii* C735 genomic sequence. We generated a synthetic construct derived from the *C. immitis* sequence, rCpa2 (NCBI Accession: WAB54687.1). Next, we identified that glucan-chitin-particles or GCP were effective adjuvants that decreased pulmonary fungal burden by eliciting both Th1 and Th17 responses without inducing a nonprotective Th2-type response [18]. We also determined that recognition in murine models is attributed to C-type lectin members Dectin-1 and Dectin-2 via CARD9 signaling [19]. However, all of these studies were limited to a single species and clinical strain (*C. posadasii*, C735). The overall question is whether GCP-rCpa1 is cross-protective among both species of *Coccidioides* as well as various clinical strains of *C. posadasii*. By creating a synthetic recombinant vaccine, we speculate that we can strategically target effective antigens that do not discriminate against the *Coccidioides* clinical isolates tested herein.

In these studies, we demonstrated that GCP-rCpa1 offers protection for C57BL/6 and HLA-DR4 transgenic mice against both *C. posadasii* and *C. immitis*, despite the seven amino acid substitutions present in the proteins and peptides of the multivalent rCpa1 antigen between the two fungal species, as described in the aligned sequence between *Cp-*derived rCpa1 compared with *Ci*-derived rCpa2. The principal advantages of multivalent subunit vaccines are their safety, only containing microbe-specific proteins or synthetic peptides, a chemically defined adjuvant, and delivery system. Recombinant vaccine antigens can be engineered to overcome antigenic variability and remove human homologous sequences using sequence-based prediction tools and genomic/bioinformatic analysis. Immuno-bioinformatics analysis using a ProPred algorithm (http://crdd.osdd.net/raghava/propred/, accessed on 1 July 2020) containing information for 51 alleles of human leukocyte antigen type HLA-DR, along with privileged MHC epitope prediction software for commonly used laboratory mice and 61 human HLA alleles (IoGenetics LLC, personal communication with Dr. Jane Homan), was applied to predict potential MHC II-binding epitopes in the rCpa1 antigen. Results suggest that six of the seven substitutions are not located on the predicated MHC II-binding epitope regions of humans nor C57BL/6 mice. The substitution (W_576_ > F_576_) in the Plb-P6 epitope may not alter the immunogenic property and protective efficacy of the rCpa1 antigen for C57BL/6 mice, since it is not recognized by their MHC II molecules (I-Ab allele) [18]. Additionally, this substitution does not alter the protective efficacy of the rCpa1 antigen for C57BL/6 and HLA-DR4 mice against either *C. posadasii* or *C. immitis* infection, as shown in Figure 1, Figure 2 and Figure 5.

Successful vaccines against *Coccidioides* infection should confer protection against a broad spectrum of clinical isolates [14]. Although the two species are indistinguishable by serological testing and disease manifestation, they do vary in geographic distribution. *C. immitis* has been historically present in desert regions of Central and Southern California, USA, while *C. posadasii* is typically present outside of California (Nevada, Arizona, New Mexico, west Texas, Mexico, and Central and South America). Both species are estimated to have diverged around 5 million years ago [5,8], sharing major virulence determinants and structural genes, although comparison of their genomes has revealed some variation in aa sequences of their translated proteins [1,27]. *Cp* mycelia grows significantly faster at 37 °C compared with *Ci,* though the growth rates of parasitic spherules for those isolates at human physiological temperature have not yet been explored [36]. Clinical isolates of *Cp* and *Ci* are all highly virulent in experimental murine models of coccidioidomycosis, where LD_100_ doses for most isolates are lower than 50 viable spores [4,37]. We have evaluated the protective efficacies of the GCP-rCpa1 vaccine against three isolates of *Cp* (C735, Silveira, and RMSCC 3488) and two isolates of *Ci* (RS and RMSCC 2394). *Cp*-C735 was previously isolated from a patient who visited a VA hospital in the 1970s in San Antonio, Texas. *Cp*-Silveira was isolated from the first reported case of Valley fever from a patient who lived in Arizona in 1894 [37,38]. *Cp*-RMSCC 3488 isolate was acquired from a patient who lived in southern/central Mexico. Isolates RS and RMSCC 2394 of *Ci* were obtained from patients residing in central California and the southern California/northern Mexico regions, respectively. Overall, these five isolates present a wide geographic distribution from Arizona, California, Texas, and central Mexico [39], and were obtained at various timepoints during the course of history of the disease [4,27]. RMSCC 3488 and RMSCC 2394 genomes pose hybrid DNA sequences from two reference genome databases of *Cp* C735 and *Ci* RS, while they are typed to be *Cp* and *Ci,* respectively [7]. Concurringly, DNA sequence analysis of the orthologous proteins included in the rCpa1 antigen demonstrated that RMSCC 3488 has a *Cp*-type of the vaccine antigen and RMSCC 2394 has a *Ci*-type. We did note phenotypic differences in growth between the isolates tested, particularly that *Cp*-RMSCC 3488 had a slower saprobic growth rate at 37 °C (24 h growth delay) compared with *Cp*-C735 or *Cp*-Silveira. This phenomenon has been recently described by Mead et al., where there is a slower growth rate than at an ambient temperature of 28 °C [36]. Additionally, *Ci*-RMSCC 2394 had an increased growth rate compared with the *Ci*-RS strain, however, differences in their pathogenesis have not been well characterized. Remarkably, we found vaccination with GCP-rCpa1 provided almost identical levels of protection for C57BL/6 mice against these five isolates of *Coccidioides*. In each case, the pulmonary fungal burden was significantly reduced by four log_10_ orders at 14 DPC.

Transgenic HLA-DR4 mice are engineered on a C57BL/6 genetic background and express chimeric MHC II alleles with a peptide binding specificity identical to human HLA-DRB1*04:01. We have noted several differences between HLA-DR4 and C57BL/6 mice that could contribute to differences in susceptibility to *Coccidioides*. HLA-DR4 mice have slightly lower bodyweight compared with C57BL/6 mice during the first 3 months (12 weeks) of their lives. HLA-DR4 can recognize different human-specific epitopes; particularly, HLA-DR4 mice produce significantly more IL-17a spot-forming units (SFU) to the five synthetic peptides that contain human epitopes P1, P2, and P10 compared with C57BL/6 mice [18]. However, HLA-DR4 mice have significantly reduced numbers of memory T cells that recognize rCpa1 compared with C57BL/6 counterparts (786 ± 67 versus 1038 ± 106 per 10^6^ splenocytes, respectively) [18]. In these studies, vaccination with GCP-rCpa1 also had significant reduction of fungal burden for highly susceptible HLA-DR4 mice against *Cp* and *Ci* (2–4 log_10_) at 9 DPC. Taken together, these results support our hypothesis that GCP-rCpa1 has cross-protective activity against various isolates of *Cp* and *Ci*. These data also suggest that the seven aa substitutions in the rCpa1 vaccine antigen do not impact protective efficacy against either species of *Coccidioides*.

The strongest immune correlative of protection against pulmonary coccidioidomycosis is in the early acquisition of Th1 and Th17 cells, which produce IFN-γ and IL-17 in the lungs, respectively [23,25,40]. Both Th1 and Th17 cells play a central role in defense against *Coccidioides* infection [18,19,41,42]. The genetically resistant DBA/2 mice developed a Th1-biased response with an early induction of IFN-γ, whereas susceptible BABL/c mice showed an early secretion of the Th2 cytokine IL-4 [41]. Interestingly, despite demonstrating high fungal burden in nonvaccinated mice, we were unable to detect significant levels of Th2-type cytokines in C57BL/6 mice. In a live-attenuated vaccine model utilizing ΔT, we have noted progressive expansion of Th1 and Th17 cells but limited expansion of Th2 cells by flow cytometry in vaccinated mice compared with nonvaccinated mice [23]. Furthermore, previous studies evaluating the immunogenicity of GCP-rCpa1 vaccine via cytokine recall assay demonstrated that following at least a 2-week resting period, the mock immunized group (GCP-MSA) was unable to elicit nonspecific T-helper response. Future studies will allow at least 3–4 weeks of resting period to prevent nonspecific T-helper response observed by the GCP adjuvant. GCP-rCpa1-immunized mice elicited a strong mixed Th1/Th17 response with little Th2 response by IL-4 production [18]. Further studies will be needed in order to characterize the role of Th2 cells in coccidioidomycosis. Neutralization of IL-12, a critical cytokine for differentiation of Th1 cells, in BDA/2 mice led to increased susceptibility to *Cp*-Silveira infections [41]. Additionally, C57BL/6 and BALB/c mice are able to produce IL-10 at greater levels compared with DBA/2 mice. Many studies have shown IFN-γ production as a correlate of vaccine-induced protection for mice against *Coccidioides* infection [4,43,44,45,46]. An attenuated live vaccine (ΔCps1) created from *Cp*-Silveira isolate elicits a Th1-biased response that confers protection against pulmonary challenge with the autologous isolate or *Ci*-RS isolate [31,37]. In humans, polymorphisms in the IL-12/IFN-γ signaling pathway result in STAT1 gain-of-function mutations that are associated with increased disease severity in *Coccidioides* infection [47]. A homozygous C_186_Y mutation in the IL-12β1 receptor is associated with increased risk of disseminated coccidioidomycosis [48]. Furthermore, a case report shows that supplementation of antifungal agents with IFN-γ slowed disease progression, and the addition of IL-4 and IL-13 blockade with dupilumab resulted in rapid resolution of a patient’s clinical symptoms [49]. Those observations support the hypothesis that Th1 response plays an important role in defense against *Coccidioides* infection.

We have previously reported that subcutaneous vaccination of IFN-γ- and IL-4-receptor knockout mice with an attenuated, live vaccine (ΔT) derived from *Cp-*C735 isolate elicited protective immunity against pulmonary challenge with an autologous virulent isolate [49]. In contrast, fungal burden, clearance, and survival are significantly compromised in mice defective in IL-17a or IL-17r [23]. Similarly, we demonstrated that depletion of IL-17a in mice reduces the protective efficacy of the GCP-rCpa1 vaccine against a pulmonary challenge with *Cp*-C735 isolate [18,19]. A recent patient with disseminated coccidioidomycosis was found to have a STAT3 mutation. STAT3 mediates IL-23 signaling, which is critical for IFN-γ, IL-12, and IL-17 production [23,50]. These observations suggest that Th17 cells and IL-17 are indispensable for vaccine immunity against *Coccidioides* infection.

Both Th1 and Th17 cells are significantly elevated against *Cp*-C735 infection at 7 DPC, as previously reported (Figure 3A; [18,19]). Similarly, Th1 and Th17 cells are also increased in response to a pulmonary challenge with isolates *Ci*-RS and *Ci*-2394 at 7 DPC (Figure 4). Notably, only Th17 cells are significantly increased in mice vaccinated against isolates *Cp*-Silveira and *Cp*-3488 at 7 and 14 DPC, respectively (Figure 3B,C). The GCP-rCpa1-vaccinated mice induce activation of pulmonary Th1 and Th17 cells at various levels and temporal patterns in response to a pulmonary challenge with various isolates of *Cp* and *Ci*. These differences are not associated with a respiratory challenge with *Cp* versus *Ci*, though evaluation of only five isolates limits the capacity for interspecies comparisons. The most common correlate of GCP-rCpa1 vaccine-induced protection against all five isolates is significantly increased acquisition of Th17 cells in the lungs. The vaccine-induced response may be dependent on the fungal isolates, challenge dose, and host variation in an immune response. Relative contribution of Th1 and Th17 cells for mice against different *Coccidioides* isolates requires further investigation.

In summary, our study supports the concept that a combination of genetic fusion techniques to create a multivalent antigen is a prospective approach for the generation of a broadly effective vaccine candidate that protects against both *Cp* and *Ci*. It is imperative to investigate multivalent vaccine formulations with broad-spectrum protection and sufficient safety in the development of a human vaccine against an orphan disease such as coccidioidomycosis, which is caused by two species of fungi.

## 5. Patents

The US patent of rCpa1 (11,413,336 B2) has issued on 16 August 2022.

## Figures and Tables

**Figure 1 vaccines-12-00067-f001:**
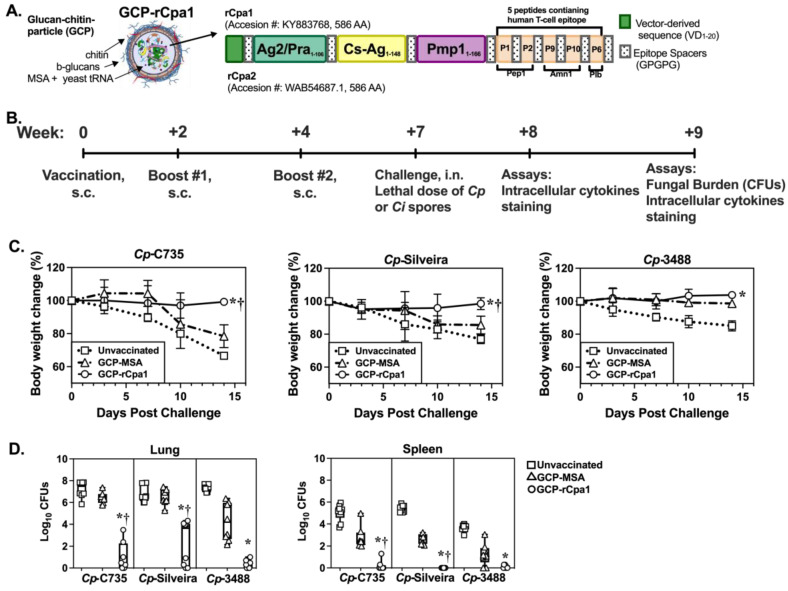
GCP-rCpa1 vaccination results in protection against multiple clinical isolates of *Cp*. (**A**) GCP-rCpa1 vaccine construct illustration. (**B**) Vaccination and immunological analysis timeline. (**C**) The daily bodyweight change (%) of C57BL/6 mice that were subcutaneously vaccinated with either the GCP-rCpa1 vaccine (circle) or GCP-MSA mock (triangle), or unvaccinated (square), then separately challenged with approximately 90–120 viable spores isolated from one of the three *Cp* clinical isolates (C735, Silveira, or 3488; representative of two independent studies, *n* = 8 mice per group). Mouse bodyweight at the time of challenge was set as 100%. Data points represent the average of percentage bodyweight measured daily for 14 days. Error bars represent the ±SD per timepoint. (**D**) Colony-forming units (CFUs) of whole tissue (lung and spleen) were determined by plate culture homogenates of unvaccinated, mock, and vaccinated mice at 14 DPC (*n* = 8 per group). The data are presented by whisker box plots of log_10_ CFUs detected from plated cultures. Asterisks and daggers indicate statistically significant differences between CFU values of the GCP-rCpa1-vaccinated versus unvaccinated mice (*, *p* < 0.05) and the vaccinated versus mock groups (†, *p* < 0.05), as determined by ordinary one-way ANOVA comparing percent weight change at 14 DPC. Statistical differences in fungal burden were conducted using Kruskal–Wallis multiple comparison tests.

**Figure 2 vaccines-12-00067-f002:**
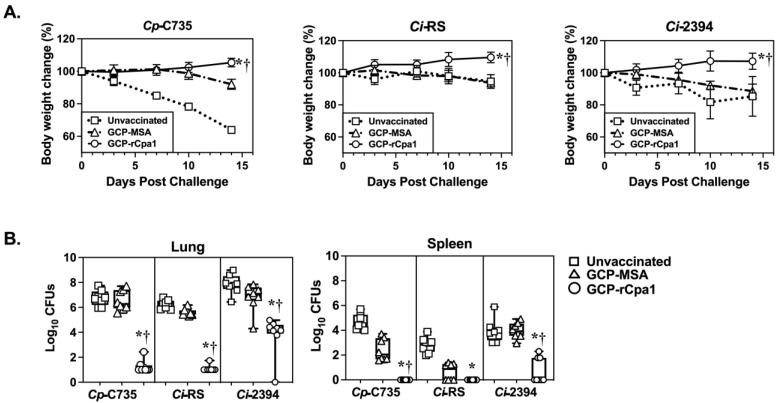
The GCP-rCpa1 vaccine confers cross-protection against isolates of *Ci*. C57BL/6 mice were vaccinated with the GCP-rCpa1 vaccine (circle), mock immunized with GCP-MSA (triangle), and unvaccinated (square) using the same schedule shown in Figure 1B. (**A**) Representative daily bodyweight changes (%) of two independent studies using the three groups of mice that were separately challenged with approximately 90–120 spores isolated from two *Ci* clinical isolates (RS and 2394). Mouse bodyweight at the time of challenge was set as 100% (*n* = 8 mice per group). Data points represent the average of percentage bodyweight measured daily for 14 days. Mice challenged with *Cp*-C735 served as a reference for comparison. (**B**) CFUs were determined by plate culture of lung and spleen homogenates of unvaccinated, mock, and vaccinated mice at 14 DPC (*n* = 8 mice per group). The data are presented as whisker box plots of CFUs (log_10_). Unvaccinated mice that were challenged with *Ci*-RS presented significantly reduced CFUs in the lungs and spleen compared with those challenged with isolate *Cp*-C735. These data suggest that *Ci*-RS is less virulent. All groups of GCP-rCpa1-vaccinated mice had reduced CFUs in the lungs and spleen compared with the mock groups. Asterisks and daggers indicate statistically significant differences between CFU values of the GCP-rCpa1-vaccinated versus nonvaccinated mice (*, *p* < 0.05) and the vaccinated versus mock groups (†, *p* < 0.05), as determined by ordinary one-way ANOVA comparing percent weight change at 14 DPC. Statistical differences in fungal burden were conducted using Kruskal–Wallis tests.

**Figure 3 vaccines-12-00067-f003:**
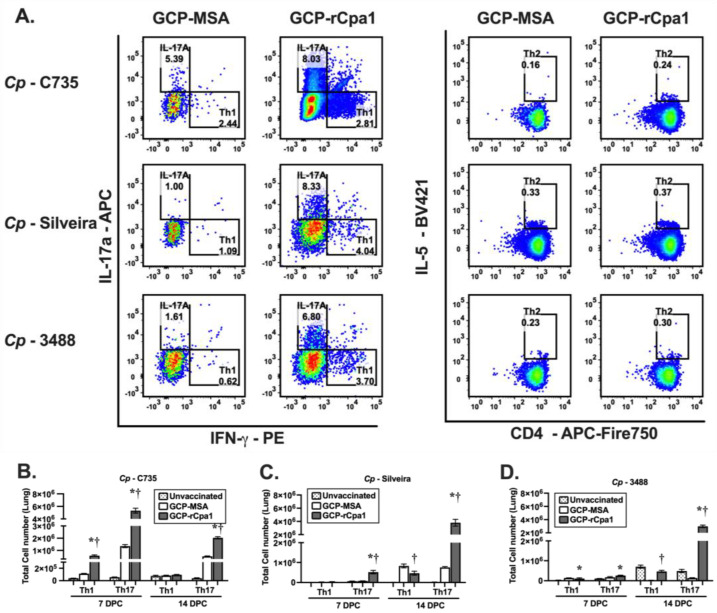
The GCP-rCpa1 vaccine induced the acquisition of Th1 and/or Th17 cells in the lungs of C57BL/6 mice that were challenged with *Cp* isolates. Pulmonary Th1, Th17, and Th2 cells expressing IFN-γ-, IL-17a, and IL-5 were evaluated using intracellular cytokine assays. The gating strategy is illustrated in (**A**). The absolute numbers (**A**–**D**) of gated, specific-cytokine-producing cells per whole lung were determined at 7 and 14 DPC. A separate group of mice was vaccinated with GCP-rCpa1 (black bars), injected with GCP-MSA (mock; white bars), or unvaccinated (dotted bars) using the protocol shown in Figure 1B. Three weeks after the final vaccination, mice were separately challenged with a lethal dose of spores isolated from *Cp-*C735 (**B**), *Cp*-Silveira (**C**), and *Cp*-3488 (**D**). Asterisks and daggers in panels B to D indicate significantly higher absolute numbers of the respective T-cell phenotypes in the lungs of the GCP-rCpa1-vaccinated versus nonvaccinated mice (*, *p* < 0.05) and the vaccinated versus mock groups (†, *p* < 0.05). Four mice per group per timepoint were used. The results are presented as mean values ±SEM and were analyzed using ordinary one-way ANOVA.

**Figure 4 vaccines-12-00067-f004:**
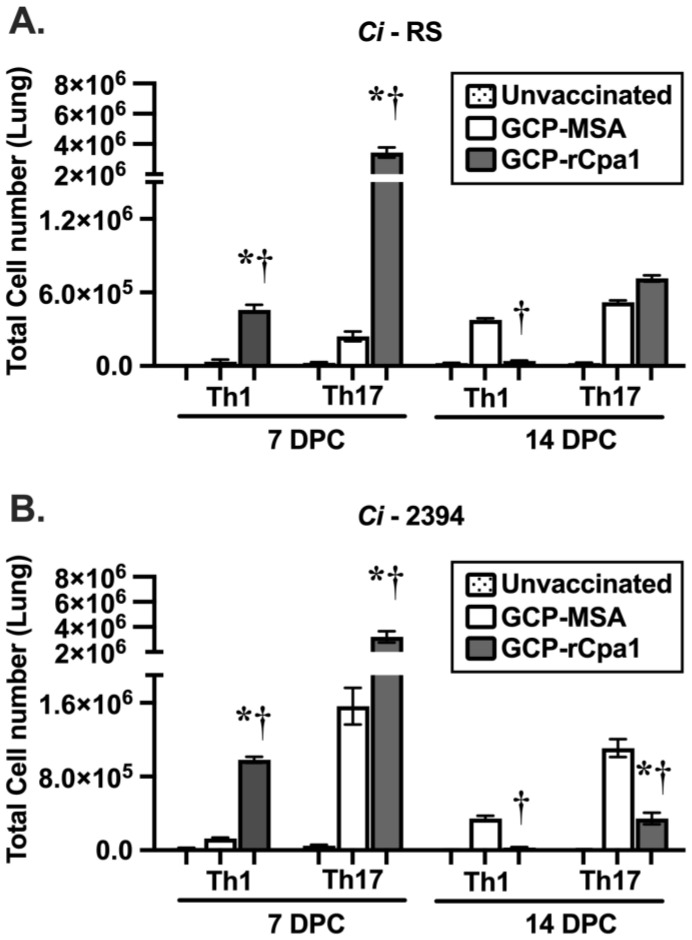
The GCP-rCpa1 vaccine induced acquisition of Th1 and/or Th17 cells in the lungs of C57BL/6 mice that were challenged with *Ci* isolates. Pulmonary Th1 and Th17 in the lungs of *Ci*-RS (**A**) and *Ci*-2394 (**B**) were enumerated, as described in Materials and Methods. Statistical significance for both Th1 and Th17 responses was observed in both *Ci* isolates (RS and 2394, (**A**,**B**)) at 7 DPC comparing unvaccinated with GCP-rCpa1 vaccinated mice. Total cell numbers of both Th1 and Th17 cells were significantly elevated in the GCP-rCpa1 vaccinated mice compared with mock and control mice at 7 DPC (*, *p* < 0.05), GCP-rCpa1-vaccinated versus unvaccinated mice (†, *p* < 0.05), and GCP-rCpa1-vaccinated versus mock mice. Four mice per group per timepoint were utilized. The results are presented as mean values ±SEM and were analyzed using ordinary one-way ANOVA.

**Figure 5 vaccines-12-00067-f005:**
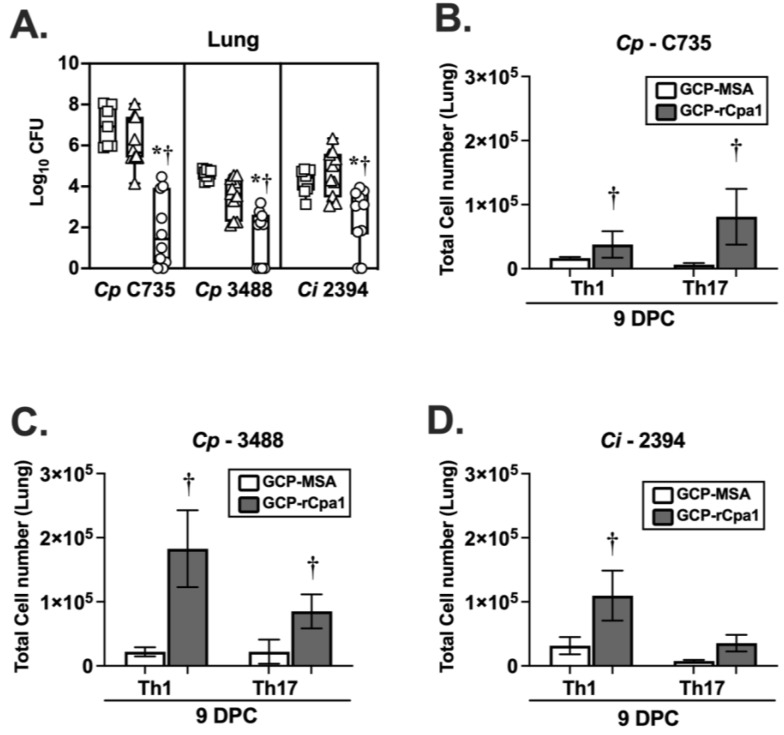
The GCP-rCpa1 vaccine elicits cross protection and a mixed Th1 and Th17 response in the lungs of HLA-DR4 transgenic mice. HLA-DR4 (DRB1*04:01 allele) transgenic mice were immunized and then separately challenged with *Cp*-C735, *Cp*-3488, and *Ci-*2394. HLA-DR4 mice were subcutaneously vaccinated with the GCP-rCpa1 vaccine (circle), mock vaccinated (triangle, GCP-MSA), or unvaccinated (square) and then separately challenged with approximately 90–120 viable spores isolated from *Cp* isolates (C735 or 3488) or *Ci* (2394) (10 mice per group). Whisker box plots presented CFU (log_10_) values from whole lungs at 9 days post challenge (**A**). Pulmonary Th1 and Th17 cells were evaluated by intracellular cytokine staining assays. Asterisks and daggers indicate statistically significant differences between CFU values of the GCP-rCpa1-vaccinated versus nonvaccinated mice (*, *p* < 0.05) and the vaccinated versus mock groups (†, *p* < 0.05). Differences in fungal burden were determined using Kruskal–Wallis tests. (**B**) Both Th1 and Th17 cells were significantly elevated for GCP-rCpa1-vaccinated mice compared with mock mice (GCP-MSA) that were challenged with *Cp*-C735 (**B**), *Cp*-3488 (**C**), and *Ci-*2394 (**D**). Ordinary one-way ANOVA was used to evaluate vaccinated versus mock groups (†, *p* < 0.05), and eight mice per group were used for flow cytometry studies.

**Table 1 vaccines-12-00067-t001:** Amino acid substitutions of coccidioidal vaccine antigens and peptides between the rCpa1 and rCpa2 constructs of *C. posadasii* (*Cp*) and *C. immitis* (*Ci*), respectively.

Antigen/Epitope	*Cp* ^a^ (39 Isolates) ^b^ rCpa1	*Ci* ^a^ (17 Isolates) ^c^ rCpa2	GenBank no. *Cp* (aa Position) ^d^	GenBank no. *Ci* (aa Position) ^d^
Ag2/Pra	D_117_	E_117_	EER27008.1 (91)	XP_001240075.1 (91)
Cs-Ag	A_164_	T_164_	AAN73410.1 (27, 41)	XP_001247410.1 (27, 41)
	P_178_	A_178_		
Pmp1	M_424_	I_424_	ABB42829.1 (136, 142, 163)	XP_001241932.1 (136, 142, 163)
	Q_430_	K_430_		
	I_451_	F_451_		
Plb-P6	W_576_	F_576_	ABA12208.1 (522)	XP_001241137.2 (522)

^a^ One letter amino acid abbreviation with a subscript number indicating position in the multivalent *Cp*-rCpa1 construct (GenBank accession AVH85517.1) and *Ci*-rCpa2 construct (GenBank accession WAB54687/1). Amino acid sequences of each antigen were aligned using a multiple sequence alignment (ClustalW) tool. ^b^ Geographic distribution of *C. posadasii* (*Cp*) isolates: California *(n =* 4), Arizona (*n* = 6), New Mexico (*n* = 1), and nondetermined (*n* = 28). ^c^ Geographic distribution of *C. immitis* (*Ci*) isolates: San Joaquin Valley*,* California (*n* = 4) and non-San Joaquin Valley (*n* = 13). ^d^ Corresponding positions of the aa substitution in the reference sequence of *Cp* or *Ci.*

## Data Availability

Data are contained within the article and Appendix A.

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
