# Peer review of "A Recombinant Multivalent Vaccine (rCpa1) Induces Protection for C57BL/6 and HLA Transgenic Mice against Pulmonary Infection with Both Species of Coccidioides"

_vaccines, 2024, doi:10.3390/vaccines12010067_

Round 1
Reviewer 1 Report
Comments and Suggestions for Authors
In this manuscript by Campuzano et al, the authors examine the cross-protective potential of a Coccidioidomycosis vaccine against infection by two species of the fungal pathogen as well as multiple strains of each species. Overall, the manuscript is well-written and the conclusions are justified. There are only minor points that the authors need to address.
- In Figure 3, the second part of 3A (the IL-5 data) is not discussed. For clarity, please either include some mention of these data or remove the IL-5 portion from the figure.
Author Response
We would like to thank the reviewer for their valuable input. We have ensured that Figure 3A and IL-5 data are addressed, as shown in the edited manuscript lines 218-219, 235 and 239-244.
In this manuscript by Campuzano et al, the authors examine the cross-protective potential of a Coccidioidomycosis vaccine against infection by two species of the fungal pathogen as well as multiple strains of each species. Overall, the manuscript is well-written and the conclusions are justified. There are only minor points that the authors need to address.
- In Figure 3, the second part of 3A (the IL-5 data) is not discussed. For clarity, please either include some mention of these data or remove the IL-5 portion from the figure.
Reviewer 2 Report
Comments and Suggestions for Authors
The authors have delivered a novel particle based adjuvant system along with a multivalent vaccine approach for the publication. They should dicuss the following reference by Galangiani et al 2022 ( doi: 10.3390/jof8080838) and the merits/demerits compared to their approach in the discussion.
Comments on the Quality of English LanguageNo comments
Author Response
We would like to thank the reviewer for their kind input; we have ensured to discuss the live attenuated vaccine work and merits/demerits under the Discussion section, lines 321-335.
The authors have delivered a novel particle based adjuvant system along with a multivalent vaccine approach for the publication. They should dicuss the following reference by Galangiani et al 2022 ( doi: 10.3390/jof8080838) and the merits/demerits compared to their approach in the discussion
Reviewer 3 Report
Comments and Suggestions for Authors
This manuscript describes a substantive investigation of the potential for a new synthetic, multivalent vaccine to confer broad protection against infection by Coccidioides species, isolates of which present genetic diversity and hybridization that complicate development of a universal antigen. Multivalent T cell antigen vaccine design is a common approach for application to other pathogens, especially when seeking to broaden the spectrum of protection against genetic variants, and it is well worthy of investigation here. The results shown here contribute important new information to advance research in this field.
Major strengths of this research include examination of sequence substitution variants in a larger number of isolates than were previously available in genomic repositories, convincing characterization of the immune response elicited, and an interesting and informative discussion section.
The limitations of animal studies are appropriately acknowledged, and would need to be addressed further in the future, but that is beyond the scope of the present work.
I suggest proofreading and several minor editorial corrections to the text, as follows.
Line 58: “candidates” (plural)
Lines 146-147: I believe the authors have used a Mann-Whitney rank sum test.
Line 170-171: “Overall, these data have validated that these 7 aa substitutions exist between Cp and Ci (Table 1).”
Line 216: ordinary one-way ANOVA
Lines 342-343: I suggest changing “and a lack of underreported illness” to “and an underreporting of the illness”.
Lines 372-374: I suggest modifying this sentence to read: “One considerable advantage of a multivalent subunit vaccine is its safety, its components including only microbe-specific proteins or synthetic peptides, a chemically-defined adjuvant, and a delivery system.”
Comments on the Quality of English LanguageThe English language style and usage are fine, though the manuscript would benefit from proofreading. In a few places the meaning of sentences was not precisely as intended due to mistakes in wording. I pointed out a few examples in comments to authors.
Author Response
We would like to thank the reviewer for their helpful input. We have made the following modifications as described by the reviewer.
- Line 58: “candidates” (plural)
We have corrected the plural candidates in line 59.
- Lines 146-147: I believe the authors have used a Mann-Whitney rank sum test.
Thank you for catching this mistake; it has been changed in line 149.
- Line 170-171: “Overall, these data have validated that these 7 aa substitutions exist between Cp and Ci (Table 1).”
We have added exist to lines 174-175.
- Line 216: ordinary one-way ANOVA
Thank you for pointing this out. We have made changes to the figure legends in Figures 1, 2 and 3.
- Lines 342-343: I suggest changing “and a lack of underreported illness”to “and an underreporting of the illness”.
We have made the suggested correction in lines 316-317
- Lines 372-374: I suggest modifying this sentence to read: “One considerable advantage of a multivalent subunit vaccine is its safety, its components including only microbe-specific proteins or synthetic peptides, a chemically-defined adjuvant, and a delivery system.”
We thank the reviewer for their suggestion and have made modifications to the original discussion section in lines 321-335 to address this suggestion and other reviewers' suggestions.
Reviewer 4 Report
Comments and Suggestions for Authors
Dear authors, I consider that the article is very well organized, with a well-designed method and taking into account quality standards and ethical norms for animal handling. What I request is that the following observations be corrected before the editor can proceed with it.
On line 32 instead of saying "Valley Fever you should say "San Joaquin Valley Fever"
On line 33: It is not correct to write (Ci) and (Cp) inside parentheses, you should write (C. immitis) and (C. posadasii)
In lines 33 and 34: It is important to specify that in the southwest of the United States of America the most prevalent species is C. immitis and in the rest of the continent it is C. posadassi. Find reference that supports this.
On line 162: the abbreviations Ci and Cp could only be placed in the table and their meaning indicated at the bottom of it.
Line 520: References: Correct references according to MDP
Author Response
We want to express our gratitude for your thorough review and providing constructive feedback. We have made the following corrections based on the reviewer's recommendations:
- On line 32 instead of saying "Valley Fever you should say "San Joaquin Valley Fever"
We have corrected line 33, as highlighted in the corrected manuscript.
- On line 33: It is not correct to write (Ci) and (Cp) inside parentheses, you should write (C. immitis) and (C. posadasii)
To avoid unnecessary repetition throughout the manuscript, we opted to abbreviate the terms C. posadasii and C. immitis as Cp and Ci, respectively. We have modified line 34-35 for clarity.
- In lines 33 and 34: It is important to specify that in the southwest of the United States of America the most prevalent species is C. immitis and in the rest of the continent it is C. posadasii. Find reference that supports this.
We thank you for your comment, we have described the geographic distribution of C. posadasii and C. immitis under the discussion on lines 378-385.
- On line 162: the abbreviations Ci and Cp could only be placed in the table and their meaning indicated at the bottom of it.
We thank you for catching this error, and have made modifications to Table 1 indicating the meaning of Ci and Cp in the figure legend and title.
- Line 520: References: Correct references according to MDP
We have modified references to MDPI style in line 514.
Reviewer 5 Report
Comments and Suggestions for Authors
The authors demonstrate recombinant vaccine efficacy to elicit protection against infection with the fungal pathogen Coccidioides. This genus comprises two human-pathogenic species that are clinically significant and are displaying expanded geographic distributions potentially arising from climate change. They previously identified a recombinant antigen from one of the species, developed an adjuvant-based vaccine, and demonstrated protection against mouse infection with a strain of the autologous species. In this study, they perform sequence analyses of the target antigens in a number of clinical isolates of both species. They test vaccine efficacy against mouse infection with several isolates of both species, measured by maintenance of body weight and limitation of fungal lung burden, and demonstrate protection against homologous and heterologous species. Further, they examine immune correlates of protection using flow cytometry to characterize host cellular responses. The manuscript is quite well written with extensive discussion and citation of relevant literature. The experimental design is solid and includes appropriate controls. The results support the authors’ interpretations and conclusions.
The formatting of Table 1 needs to be spatially reoriented, e.g., in the heading line, “Cp” is misplaced and there is erroneous spacing for “Antigen/ epitope.”
Author Response
We appreciate the reviewer's valuable feedback. We have modified Table 1 to adjust for the excess spacing and made additional corrections based on other reviewers' comments on the table and figures.
Thank you for taking the time to review and share your thoughts.
The authors demonstrate recombinant vaccine efficacy to elicit protection against infection with the fungal pathogen Coccidioides. This genus comprises two human-pathogenic species that are clinically significant and are displaying expanded geographic distributions potentially arising from climate change. They previously identified a recombinant antigen from one of the species, developed an adjuvant-based vaccine, and demonstrated protection against mouse infection with a strain of the autologous species. In this study, they perform sequence analyses of the target antigens in a number of clinical isolates of both species. They test vaccine efficacy against mouse infection with several isolates of both species, measured by maintenance of body weight and limitation of fungal lung burden, and demonstrate protection against homologous and heterologous species. Further, they examine immune correlates of protection using flow cytometry to characterize host cellular responses. The manuscript is quite well written with extensive discussion and citation of relevant literature. The experimental design is solid and includes appropriate controls. The results support the authors’ interpretations and conclusions.
The formatting of Table 1 needs to be spatially reoriented, e.g., in the heading line, “Cp” is misplaced and there is erroneous spacing for “Antigen/ epitope.”